# Unveiling the Link: Exploring Mitochondrial Dysfunction as a Probable Mechanism of Hepatic Damage in Post-Traumatic Stress Syndrome

**DOI:** 10.3390/ijms241613012

**Published:** 2023-08-21

**Authors:** Marina V. Kondashevskaya, Lyudmila M. Mikhaleva, Kseniya A. Artem’yeva, Valentina V. Aleksankina, David A. Areshidze, Maria A. Kozlova, Anton A. Pashkov, Eugenia B. Manukhina, H. Fred Downey, Olga B. Tseilikman, Oleg N. Yegorov, Maxim S. Zhukov, Julia O. Fedotova, Marina N. Karpenko, Vadim E. Tseilikman

**Affiliations:** 1A.P. Avtsyn Research Institute of Human Morphology, B.V. Petrovsky National Research Center of Surgery, Moscow 119991, Russiamikhalevalm@icloud.com (L.M.M.);; 2Scientific and Educational Center ‘Biomedical Technologies’, School of Medical Biology, South Ural State University, Chelyabinsk 454080, Russia; 3Federal Neurosurgical Center, Novosibirsk 630048, Russia; 4Department of Physiology and Anatomy, University of North Texas Health Science Center, Fort Worth, TX 76107, USA; 5Institute of General Pathology and Pathophysiology, Moscow 125315, Russia; 6Faculty of Basic Medicine, Chelyabinsk State University, Chelyabinsk 454080, Russia; 7Laboratory of Neuroendocrinology, Pavlov Institute of Physiology, Saint Petersburg 199034, Russia; 8Department of Physiology, Pavlov Institute of Experimental Medicine, Saint Petersburg 197376, Russia; 9Zelman Institute of Medicine and Psychology, Novosibirsk State University, Novosibirsk 630090, Russia

**Keywords:** post-traumatic stress disorder, anxiety, mitochondria, rats, liver, hepatocytes, cytokines, inflammation, oxidative stress, phenotypes

## Abstract

PTSD is associated with disturbed hepatic morphology and metabolism. Neuronal mitochondrial dysfunction is considered a subcellular determinant of PTSD, but a link between hepatic mitochondrial dysfunction and hepatic damage in PTSD has not been demonstrated. Thus, the effects of experimental PTSD on the livers of high anxiety (HA) and low anxiety (LA) rats were compared, and mitochondrial determinants underlying the difference in their hepatic damage were investigated. Rats were exposed to predator stress for 10 days. Then, 14 days post-stress, the rats were evaluated with an elevated plus maze and assigned to HA and LA groups according to their anxiety index. Experimental PTSD caused dystrophic changes in hepatocytes of HA rats and hepatocellular damage evident by increased plasma ALT and AST activities. Mitochondrial dysfunction was evident as a predominance of small-size mitochondria in HA rats, which was positively correlated with anxiety index, activities of plasma transaminases, hepatic lipids, and negatively correlated with hepatic glycogen. In contrast, LA rats had a predominance of medium-sized mitochondria. Thus, we show links between mitochondrial dysfunction, hepatic damage, and heightened anxiety in PTSD rats. These results will provide a foundation for future research on the role of hepatic dysfunction in PTSD pathogenesis.

## 1. Introduction

Anxiety can be defined as a temporally diffused emotional state caused by potentially harmful stress [1,2,3]. Thus, stress and anxiety have intersecting behavioral and neural elements. Stress is an adaptive response programmed to maintain homeostasis in the presence of altered environmental conditions [4]. This allostatic response acts through prolonged activation of neuroendocrine regulatory systems, especially the limbic–hypothalamic–pituitary–adrenal axis (LHPA) and the hypothalamus–pituitary–adrenal axis [5]. The LHPA requires integrity of the prefrontal cortex, the hippocampus, and the amygdala. The LHPA and the hypothalamus–pituitary–adrenal axis coordinate essential behavioral, physiological, and molecular responses to psychogenic stressors [6]. Notably, the limbic areas and the hypothalamus are highly sensitive to glucocorticoids (GCs) [7]. GCs are stress-released hormones that provide resilience to various stressors, including psychogenic insults [7].

PTSD is a stress-related disorder and produces anxiety-like behaviors [8,9,10]. These behaviors develop and continue after cessation of the stressful event as a consequence of time-dependent sensitization to the stressful stimuli [2,11].

Predator stress is considered a well-validated animal model for post-traumatic stress disorder (PTSD) [11,12,13,14,15]. Following PS exposures, rats can be segregated into high anxiety (HA) and low anxiety (LA) phenotypes [16]. This is consistent with the well-known fact that only a fraction of individuals exposed to severe stress develops PTSD. Previously, it was shown that HA rats exhibited altered monoamine metabolism in various brain areas, adrenal insufficiency, histomorphological signs of metabolic and hypoxic damage to cardiomyocytes, and impaired myocardial contractility, whereas LA rats did not show similar changes [15,16,17].

Mitochondria play a vital role in cellular homeostasis. Indeed, over 90% of cellular energy generation takes place in the mitochondria. In addition, mitochondria have important biosynthetic activities, control intracellular Ca^2+^ metabolism and signaling, regulate thermogenesis, generate most cellular reactive oxygen species (ROS), and serve as the regulator of the cell for programmed cell death (apoptosis) [18,19,20]. Given their crucial role in cell physiology, it seems obvious that mitochondria would be among the first responders to stress that challenges cellular and organismal homeostasis. Most of the primary mediators of the stress response, including hormones (GCs and catecholamines), immune factors (cytokines), and heat-shock proteins, exert numerous effects on mitochondrial biogenesis, metabolism, ROS generation, and apoptosis [21]. Mitochondria are highly sensitive to GCs, and GC receptors are found in mitochondria of several cell types [22].

Mitochondria are multifunctional, life-sustaining organelles that represent a potential, subcellular intersection point between a psychosocial experience and the biological stress response [21]. Recent studies revealed new roles of mitochondria within the *nucleus accumbens* (NAc), where they are involved in defining individual trait anxiety and participate in linking stress and anxiety [23]. Moreover, oral administration of a mitochondria-targeted antioxidant decreased anxiety-like behavior in inbred mice selected for high-anxious traits [22].

Considering the unique role of mitochondria in supporting cell viability, it is quite logical to investigate the role of mitochondrial dysfunction in the pathogenesis of stress-related illnesses, including PTSD. Notably, animal models of PTSD have shown mitochondrial dysfunction, e.g., dysregulation of oxidative phosphorylation and other metabolic pathways, including β-oxidation of fatty acids and the tricarboxylic acid pathway [24]. These PTSD models generated neural reactive oxygen species that damage DNA, proteins, and lipids [25]. Mitochondrial structure and replication were altered, and neuroinflammatory responses, signal transduction, and apoptosis were affected. With regard to PTSD, all occurrences of mitochondrial dysfunction were found in multiple brain areas. However, mitochondrial dysfunction in other organs, particularly in the liver, was not investigated.

The crucial role of the liver in maintaining homeostasis is well recognized [26,27] since it performs numerous vital functions, including glycogen storage, protein synthesis, production of very low density lipoprotein (VLDL) particles and other components of bile, and metabolism of nutrients and xenobiotics [28,29]. Glucose is stored as glycogen in the cytoplasm of the liver cells, where it serves as the main depot source for maintaining glucose homeostasis in the blood [30]. The brain uses about 69% of the glucose released by the liver for the synthesis of high-energy phosphates and neurotransmitters [31,32]. In fact, most glucose consumed by the brain is of hepatic origin [33]. Additionally, since lipid synthesis in the brain is limited, lipoproteins derived from the liver supply the brain with essential lipids [34,35].

In addition to its role as a metabolic center, the liver has recently attracted attention due to its function in the liver–brain axis, reflecting a close interaction of the liver with the central nervous system (CNS) via the autonomic nervous system [36]. Conversely, neuronal signals from the CNS influence glucose, lipid, and protein metabolism in the liver [37]. The liver communicates with important regulatory centers in the brain, including the hypothalamus, via afferent and efferent fibers of the sympathetic and parasympathetic nervous systems. Hypothalamic anorexigenic and orexigenic peptides signal the liver via neuronal networks to modulate hepatic lipid content and the production of VLDL, e.g., VLDL cholesterol [38]. In addition, peripheral hormones, including insulin, leptin, and glucagon-like-peptide-1, exert control over liver lipids by acting directly in the CNS or via peripheral nerves [39].

However, until now, the importance of the liver in supporting allostatic responses to stress has received scant attention, although recently, it has been shown that the liver is involved in regulating LHPA activity [5]. The liver affects the brain by metabolizing hormones involved in the neuro-endocrine regulation of neuronal activity. GCs are secreted by the adrenal cortex under the control of the hypothalamic–pituitary–adrenal axis, one of the main neuro-endocrine systems [40]. GCs are metabolized by cytochrome P450 (CYP) and 11-β-hydroxysteroid dehydrogenase type 1 (11βHSD1) in the liver [41].

Anxiety disorders are frequently seen in patients with liver disease [42], and it is noteworthy that an anxiety index (AI) measured in the elevated-plus-maze test (EPM) is strongly correlated with hepatic 11βHSD1 and CYP3A activities [43]. 11βHSD-1-dependent GC metabolism dominates in HA rats, while CYP3A metabolism dominates in LA rats [43]. Mechanisms linking anxiety and cardiovascular damage in humans include the development of atherogenic dyslipidemia, hypertension, and insulin resistance [27]. It is quite possible that atherogenic dyslipidemia is associated with liver dysfunction [27]. In turn, the liver is susceptible to various pathophysiological insults, including stress. In recent experimental studies, liver damage and hepatic inflammation were shown to be associated with acute and chronic psychological stress [27,28]. In the liver, stress causes necrotic lesions with perivascular and lobular infiltration of mononuclear cells and increased free radical oxidation [44]. We propose here that mitochondrial dysfunction in hepatocytes is critical in this important pathogenetic chain of events leading to liver dysfunction. Hence it is justifiable to expect that liver mitochondrial dysfunction has a direct relationship with the pathogenesis of PTSD.

## 2. Results

### 2.1. Effects of Predator Stress on Anxiety in an Elevated-Plus-Maze Behavioral Test

To evaluate the effect of different anxiety phenotypes on the liver, we first exposed rats to PS for 10 days. Then, after 14 days of rest, we performed an elevated-plus-maze (EPM) test to evaluate each rat’s level of anxiety. Based on the rat’s performance on the EPM behavioral test, its AI was calculated [see Section 4]. Based on the AI, rats were segregated into LA (AI < 0.8) and HA (AI ≥ 0.8) groups [26].

Table 1 shows the results of the behavioral test for control rats and for the PS-exposed rats after their segregation into HA and LA groups. For HA rats, the time spent in the open arms of the maze was 96% shorter than for control rats and 93% shorter than for LA rats. The LA and control rats spent similar time in the open arms and in the closed arms. As would be expected from the time spent in the open arms, the HA rats spent more time in the closed arms than the control rats (+71%) or the LH rats (+28%). The number of entries into the open arms by the HA rats was 83% less than for the control rats and 80% less than for the LA rats. LA and control rats had similar numbers of entries into the open and closed arms. The number of entries into the closed arms by HA rats was 40% less than by control rats and tended to be less (33%) than by LA rats. Since the rats were segregated into HA and LA groups, the AI of the HA rats was clearly greater than that of the LA rats. However, it is important to note that the AI of the HA rats was 55% greater than that of the control rats and 13% greater than that of the LA rats. The AIs of LA and control rats were similar. Thus, these data show marked differences in EPM test values between HA and LA rats, and thus, they confirmed that the AI was a valid criterion for segregating the rats into the HA and LA phenotypes.

### 2.2. Effects of Predator Stress on Plasma Corticosterone Concentration

The effect of PS on circulating plasma corticosterone (Cort) concentrations in rats is shown in Figure 1. In the HA group, plasma Cort concentration was 59% lower than in control rats and 33% lower than in the LA group. Plasma Cort concentration in the LA group was 39% lower than in the control group. Overall, these data demonstrate that the HA rats clearly had lower plasma Cort.

### 2.3. Effects of Predator Stress on Plasma and Hepatic Cytokines

The data presented in Table 2 show significant differences between the HA and control rats for plasma and liver concentrations of the cytokines IL-1β, IL-2, IL-4, IL-6, and IL-10. In nearly all cases, the respective concentrations in HA rats differed significantly from those in LA rats. In some cases, the respective concentrations in LA rats were not significantly different from those in control rats.

Specifically, plasma IL-1β (Figure 2E) was 73% higher in HA rats than in control rats and tended to be higher (21%, *p* < 0.1) than in LA rats. Plasma IL-1β tended to be higher (28%, *p* < 0.1) in LA rats than in control rats. Liver IL-1β was 332% higher in HA rats than in control rats and 239% higher than in LA rats (Figure 2F). Liver IL-1β in LA rats was not significantly different from that in control rats.

Plasma IL-2 was 73% higher in HA rats than in control rats and 47% higher than in LA rats (Figure 2G). Plasma IL-2 in LA rats was not significantly different from that in control rats. Liver IL-2 was 63% higher in HA rats than in control rats and 35% higher than in LA rats (Figure 2H). Liver IL-2 in LA rats was 18% higher than in control rats.

Plasma IL-4 was 12% lower in HA rats than in control rats and 20% lower than in LA rats (Figure 2C). Plasma IL-4 in LA rats was 10% higher than in control rats. Liver IL-4 was 78% lower in HA rats than in control rats and 19% lower than in LA rats (Figure 2D). Liver IL-4 was 11% higher in LA rats than in control rats.

Plasma IL-6 was 172% higher in HA rats than in control rats and 26% higher than in LA rats (Figure 2A). Plasma IL-6 in LA rats was 120% higher than in control rats. Liver IL-6 was 220% higher in the HA rats than in the control rats and 24% higher than in the LA rats (Figure 2B). Liver IL-6 of LA rats was 160% higher than in control rats.

Plasma IL-10 was 19% higher in HA rats than in control rats and 26% lower than in LA rats (Figure 2I). Plasma IL-10 was 44% higher in LA rats than in control rats. Liver IL-10 was 35% lower in HA rats than in control rats and 18% lower than in LA rats (Figure 2J). Liver IL-10 in LA rats was 21% lower than in control rats. Overall, the current data indicated an imbalance between cytokines in the liver of the HA rats. Hepatic IL-1, IL-6, and IL-2 concentrations were increased, whereas IL-4 and IL-10 concentrations were decreased.

### 2.4. Evaluation of Oxidative Stress in the Liver

Data presented in Figure 3 are indicative of the presence of oxidative stress in both stressed rat phenotypes. PS significantly decreased hepatic superoxide dismutase (SOD) activity in HA and LA rats (Figure 3A). Hepatic SOD activity in HA rats was 38% lower than in control rats and 14% lower than in LA rats. SOD activity in LA rats was 29% lower than in control rats. PS significantly increased hepatic ketodienes and conjugated trienes in HA rats (Figure 3B). In HA rats, these lipid peroxidation (LP) products were 94% higher than in control rats and tended to be higher (27%, *p* < 0.1) than in LA rats. PS significantly increased the hepatic mitochondrial content of ketodienes and conjugated trienes only in HA rats (Figure 3C). In the mitochondria of HA rats, these dienes were 24% higher than in control rats and 19% higher than in LA rats. The concentrations of these dienes were similar in LA and control rats. Overall, oxidative stress in the stressed rats was evident in the decreased hepatic SOD activity with simultaneously increased LP products. These effects were more pronounced in the HA rats.

### 2.5. PS-Mediated Increases in Aspartate Aminotransferase (AST) and Alanine Aminotransferase (ALT) Activities in Rat Liver and Blood

Figure 4 shows transaminase activities in the plasma and in the liver in both phenotypes of stressed rats. Hepatic AST activity increased markedly in HA rats (Figure 4A). AST activity was 68% higher in HA rats than in control rats and 77% higher than in LA rats. Hepatic AST activity of LA rats was similar to that of the control rats. Plasma AST activity was 42% higher in HA rats than in control rats and 35% higher than in LA rats (Figure 4C). Plasma AST activity of LA rats did not differ significantly from that of control rats.

Although hepatic ALT activity did not differ among the groups (Figure 4B), plasma ALT activity was significantly increased in HA rats. Plasma ALT was 270% higher in HA rats than in control rats and 140% higher than in LA rats (Figure 4D). Plasma ALT activity of LA rats did not differ significantly from that of control rats. Overall these data indicated an increase of transaminase activities in the plasma and liver of the HA rats that generally exceeded those in the LA rats.

### 2.6. Effects of Predator Stress on Liver Weight and Consistency

Figure 5 shows differences in the liver weight and liver index of the control and experimental rats. Macroscopically, the livers of HA rats were larger than those of control and LA rats. Liver weight was 16% greater in HA than in control rats and 13% greater than in LA rats (Figure 5A). The liver weight of LA rats did not differ significantly from that of control rats. The liver weight index was 20% greater in HA rats than in control rats and 15% greater than in LA rats (Figure 5B). This index was similar in LA and control rats. The liver of HA rats was more easily damaged when grasped with forceps, and also it had a friable texture and a grayish-brown color. All these signs are manifestations of diffuse changes in the liver of HA rats, while the liver of LA rats was macroscopically indistinguishable from the control in color and consistency. Overall, only the HA rats had increased liver weight and liver index.

### 2.7. Effects of Predator Stress on the Ultrastructure of Mitochondria

Table 2 summarizes morphometric data of rat mitochondria in the control and experimental rats. Representative ultrastructural changes in the hepatocytes of control rats and stressed rats are shown in Figure 6. In both the control group and the two experimental groups, the hepatocytes were characterized by a complete set of organelles characteristic of liver cells (Figure 6). There were no changes in their structure; only an increase in the number of mitochondria in the stressed rats was noticeable. In the hepatocytes of all rats, mitochondria were of different sizes, mostly round, occasionally oval.

Electron micrographs showed that there were much fewer mitochondria in the hepatocyte of the control rats (Figure 6A) than in the liver cells of the stressed rats (Figure 6B,C). In both HA and LA rats, the total number of mitochondria was 116% and 137% higher than in the control group (Table 2). Mitochondria typing revealed that, compared to control rats, the number of small mitochondria increased by 279% in HA rats and 117% in LH rats. The number of small mitochondria in HA rats was 73% higher than in LA rats. In contrast, the number of medium mitochondria increased by only 71% in HA rats and, to a larger degree, by 139% in LA rats. The number of medium mitochondria in LA rats was 40% higher than in HA rats. The number of large mitochondria was similar in all three groups.

Although the numbers of hepatic mitochondria generally increased in the stressed rats, their area generally decreased (Table 2). The small mitochondria area of HA rats was 37% less than in control rats and 42% less than in LA rats. Small mitochondria areas of LA and control rats were similar. The medium mitochondria area of HA rats was 54% less than the control and 13% less than LA rats. The medium mitochondria area of LA rats was 47% less than control rats. The large mitochondria area of HA rats was 65% less than the control and 31% less than LA rats. The large mitochondria area of LA rats was 65% less than control rats. Overall, these data demonstrated the prevalence of small mitochondria in the hepatocytes of the HA rats.

### 2.8. Effects of Predator Stress on Liver Histochemistry

The histochemical changes in the livers of the control and experimental rats are shown in Figure 7, Figure 8 and Figure 9. Figure 7 shows that glycogen was reduced in liver sections of rats exposed to PS, and this reduction was clearly greater in the liver of the HA than in LA rats.

Figure 8 shows liver sections stained with Sudan III for lipids. The liver of a control rat had no evidence of lipid droplets (Figure 9A). The liver of a LA rat had some lipid droplets (Figure 9B), whereas the liver of a HA rat had an intense accumulation of lipid droplets (Figure 9C).

Figure 9 shows liver sections stained with bromophenol blue for protein content. Clearly, the protein content was increased in the stressed rats and, to a greater extent, in the HA rats.

Figure 10 shows the optical density (OD) of staining as measured on the microphotographs, specifically for lipids (Sudan III), glycogen (PAS reaction), and protein (bromophenol blue). For lipids, the OD of HA rat sections was 177% greater than in the control sections and 48% greater than in the LA sections. The lipid OD of the LA sections was 26% greater than in the control sections. For glycogen, the OD of HA rat sections was 26% less than in the control sections and 16% less than in the LA sections. The glycogen OD of the LA sections was 14% less than in the control sections. For protein, the OD of the HA rat sections was 36% greater than in the control sections and tended to be greater (10%) than in the LA sections. The protein OD of the LA sections was 23% greater than in the control sections.

Overall, the data in these figures show that the accumulation of lipids and proteins and the depletion of glycogen were more pronounced in the livers of HA rats.

### 2.9. Effects of Predator Stress on Glycemia and Lipidemia

Figure 11 shows the differences between the control and both experimental groups for blood glucose, triglycerides, and cholesterol. The blood glucose in HA rats was 20% lower than in control rats and 22% lower than in LA rats (Figure 11A). Blood glucose in LA and control rats were similar (blood triglycerides were 235% higher in HA rats than in control rats and 57% higher than in LA rats (Figure 11B). Blood triglycerides in LA rats were 34% higher than in control rats. Blood cholesterol in HA rats was 33% higher than in control rats and 20% higher than in LA rats (Figure 11C). Blood cholesterol in LA and control rats was similar. Blood LDL-cholesterol was 31% higher in HA rats than in control rats and tended to be higher (13%) than in LA rats (Figure 11D). Blood LDL-cholesterol was 23% higher in LA rats than in control rats. Blood HDL-cholesterol in HA rats was 32% lower than in control rats and tended to be lower (20%) than in LA rats (Figure 11E). Blood HDL-cholesterol was similar in LA and control rats.

Overall, these data indicate that the HA rats had the lowest blood glucose and the highest blood triglycerides and cholesterol. Moreover, in the HA rats, cholesterol was redistributed from HDL to LDL.

### 2.10. Effects of Predator Stress on Ceruloplasmin Concentration in Plasma

Figure 12 shows plasma ceruloplasmin in the control and experimental rats. Plasma ceruloplasmin was 45% higher in HA rats than in control rats and 38% higher than in LA rats. Plasma ceruloplasmin in the LA rats and control rats was similar. Thus, only in the HA rats was an increase in plasma ceruloplasmin observed.

Overall, the results showed that the AI was a valid criterion for segregating the rats into the HA and LA phenotypes. Plasma Cort decreased in HA rats, and hepatic IL-1, IL-6, and IL-2 increased. IL-4 and IL-10 decreased. Oxidative stress increased in HA rats, as evidenced by decreased hepatic SOD activity and increased LP products. Transaminase activities in plasma and liver were higher in HA rats. Liver weight was increased only in the HA rats. Small mitochondria predominated in the hepatocytes of HA rats. The accumulation of lipids and proteins and the depletion of glycogen were greater in the livers of HA rats. Blood glucose was lower, and blood triglycerides and cholesterol were higher in the HA rats. Cholesterol was redistributed from HDL to LDL cholesterol in the HA rats. Plasma ceruloplasmin increased only in the HA rats. Thus, the results show links between mitochondrial dysfunction, hepatic damage, and heightened anxiety in PTSD rats.

## 3. Discussion

Previously, we reported the role of the liver in the allostatic response to experimental PTSD [43]. Liver damage in PTSD rats has also been recently reported [45]. Here, for the first time, we have demonstrated mitochondrial dysfunction in the liver of experimental PTSD rats. Until now, mitochondrial dysfunction has been demonstrated only in the brain [24]. Analysis of the current results allows us to suggest probable mechanisms of the observed damage of liver and hepatic mitochondrial damage and to discuss the consequences of the subsequent hepatic mitochondrial dysfunction.

The main findings of this study are: First, experimental PTSD induced by exposure of rats to PS caused liver damage that was significantly greater in the HA phenotype rats. Such pronounced liver damage did not occur in similarly stressed LA phenotype rats. Second, in the HA rats, liver damage and anxiety were associated with an imbalance between GCs and cytokines that expressed in the decrease of blood Cort with a simultaneous increase of IL-1 and IL-2 and with a decrease of IL-4 and IL-10 concentrations in the blood and in the liver. IL-6, an acute-phase mediator, was also increased in the blood and liver, significantly more so in HA rats. Third, liver damage in the HA rats was associated with oxidative stress. Oxidative stress was expressed by the decrease in hepatic SOD activity with a simultaneous increase in the content of LP products. The resulting liver damage was reflected in lipid dystrophy, glycogen depletion, and a reduction in mitochondrial area. Fourth, liver damage in HA rats was associated with hypoglycemia and dyslipidemia.

Figure 13 illustrates a positive feedback loop that we propose to account for the liver damage induced by stress in the HA phenotype. The results of the current investigation agree with each component of this loop.

The ability of GCs to inhibit the synthesis, release, and/or the potency of cytokines that promote inflammatory responses has traditionally been considered to have significant endocrine effects during stress [46]. GC deficiency has been associated with pathological overshoots of inflammatory responses. Usually, chronic stress is thought to increase plasma Cort concentration [7,46]; however, this PS model of PTSD is characterized by elevated GC metabolism in the liver due to an increase of 11βHSD1 activity [43]. Consequently, plasma Cort was reduced in the PS rats and significantly more so in the HA rats than in the LA rats.

Previously, we reported that chronic PS leads to a decrease in plasma Cort with a concomitant increase in proinflammatory cytokines [17,43]. These imbalances support the development of inflammation. Currently, neuroinflammatory responses are considered potential risk factors for anxiety disorders [46]. The positive correlations between the plasma concentrations of IL-1, IL-2, IL-6, and AI and the negative correlation between the plasma concentration of Cort and AI (Figure 14) are in good agreement with prior reports.

GC deficiency might mediate the development of liver damage since GCs at low concentrations are unable to downregulate the effects of proinflammatory cytokines on hepatocytes. This notion is supported by positive correlations between liver concentration of IL-1, IL-2, and Sudan III stain in liver sections of stressed rats. Accumulation of fat in the liver may lead to inflammation and tissue damage. Budick-Harmelin et al. reported that elevated triglycerides potentiate an inflammatory response in rats’ Kupfer cells [47]. On the one hand, fat accumulation in the liver can result from increased production, but on the other hand, this can result from reduced secretion due to reduced protein production. However, the liver protein was not reduced in the PS rats. To what extent this increase in protein reflects an acute phase response secondary to the elevation of IL-6 merits further investigation. However, it is notable that in the liver of HA rats, there was a large increase in IL-6, and IL-6 is considered an acute-phase mediator. Correspondingly, in the HA rats, there was an increased concentration of plasma ceruloplasmin. Ceruloplasmin was positively correlated with plasma IL-6 as well as with bromophenol blue in the stained sections of the liver (Figure 14).

C-reactive protein, alpha-1-antitrypsin (A1AT), alpha-1 glycoprotein, haptoglobin, and C3c and C4 complement fractions are considered acute-phase proteins [48]. Moreover, it has been suggested that IL-6 may represent the major mediator of acute-phase proteins [49]. Recently it has been reported that increased IL-6 concentration in the liver is associated with an imbalance between Th17/regulatory T (Treg) cells [50]. Importantly, apart from IL-6 in the liver, there were increased concentrations of IL-1 and IL-2, along with a simultaneous decrease in IL-4 and IL-10. Meanwhile, IL-10 is considered a marker of Treg cells [50]. It is likely that the simultaneous increase of IL-6, along with the decrease of IL-10 in HA rats, reflects a disturbance of the hepatic T-helper/T-suppressor ratio. This hypothesis will be tested in future research. It is quite possible that the observed decrease in hepatic IL-10 was restricted to the liver, whereas the increase in plasma IL-10 reflected an increase of this cytokine in other organs. Interestingly, plasma IL-10 was increased in PTSD patients with comorbid depression [51].

With regard to inflammatory biomarkers, the results of this study are in good agreement with a recent systematic review regarding inflammation in PTSD [52]. Their attention was focused on specific inflammatory biomarkers and on the potentially harmful effects of decreased ROS clearance and of reduced antioxidant activity, including reduced CAT, glutathione peroxidase (GPX), and SOD activities. Although IL-1 and IL-6 were considered proinflammatory in that review, the term “proinflammatory cytokines” applied to them was incorrect [53]. Indisputably, these cytokines have many diverse pleiotropic effects beyond inflammation [54], including hormonal-like systemic effects [55]. In particular, IL-1 can stimulate the hypothalamic–pituitary–adrenal (HPA) axis to release GCs [56,57]. Thus, these cytokines provide a bidirectional link between the immune and nervous systems. In fact, nearly forty years ago, Blalock proposed that “the immune system also serves a sensory role, a sixth sense to detect things the body cannot otherwise hear, see, smell, taste or touch” [58].

IL-1 and IL-6 are the first signal molecules discovered to be common to the immune and neuroendocrine systems. When interpreting our results with respect to cytokines, it may still be advisable to preserve the terms proinflammatory cytokines. This may be justified, especially with regard to IL-1. In an earlier study of repeated restraint stress (RRS), RRS caused necrotic foci in the liver with perivascular and lobular infiltration of mononuclear cells [44]. There was an increase in free radical oxidation and an increase in IL-1 concentration. The IL-1 receptor antagonist Anakinra prevented the infiltration of mononuclear cells and reduced the level of free radical oxidation and necrotic lesions [44].

Our current data suggest that in HA rats, some cytokines are associated with oxidative stress. This is evident from the positive correlations between the concentration of liver IL-2 and liver LP product and between liver IL-6 concentration and mitochondrial LP product. At the same time, IL-4 and IL-10 concentrations were negatively correlated with mitochondrial LP products. These correlations (Figure 14) allow us to recognize the proinflammatory potential of IL-2 and IL-6 as well as the anti-inflammatory potential of IL-4 and IL-10.

In HA rats, oxidative stress was reflected in a decrease in hepatic SOD activity and in an increased content of ketodienes and conjugated trienes. Ketodienes and conjugated trienes have severe toxic effects, so it is most likely that they were involved in the observed hepatic damage. This view is supported by positive correlations between AST and ALT and the increased hepatic content of ketodienes and conjugated trienes (Figure 14). In addition, plasma AST and ALT are considered markers of liver hepatocellular damage [59,60].

It is well known that proinflammatory mediators may also alter mitochondrial function via an increase in mitochondrial oxidation stress. In turn, ROS production is the essential signaling step in the inflammatory response [61]. Mitoptosis is a consequence of mitochondrial ROS overproduction. Meanwhile, mitochondrial DNA (mtDNA) may be broken into smaller fragments. These degraded elements of mtDNA can induce additional increases in IL-1β, IL-6, and TNF concentrations, and thus, they form a vicious circle between proinflammatory factors and mitochondrial damage [62].

In the current study, for the first time in a model of PTSD, a change in the size of hepatic mitochondria was noted. Moreover, a large accumulation of altered liver organelles was observed in the HA rats. Mitochondrial homeostasis is normally achieved by a balance between mitochondrial biogenesis and mitophagy [63]. It has been reported that smaller mitochondria result from increased mitochondrial fission [64], and excessive mitochondrial fission is a response to hypoxia–reoxygenation stress [65].

It is well known that the number and size of mitochondria are markers of mitochondrial functional activity. Small mitochondria are characteristic of resting cells with low respiratory activity, whereas big mitochondria are typically associated with increased ATP production, higher transmembrane potential, and decreased production of reactive oxygen species (ROS) [66]. In HA rats, an increase in the number of small mitochondria predominated, whereas, in LA rats, the number of medium-sized mitochondria in the liver was significantly increased. It is noteworthy that an analysis of the phenotypic differences in rat resistance to hypoxia showed similar results [67]. The predominance of small mitochondria was shown to be characteristic of rats both with low resistance to hypoxia and with high anxiety [67]. Perhaps the most intriguing result of this study is the positive correlation between the intensity of hepatic lipid degeneration and the number of small mitochondria in stressed rats (Figure 14). Moreover, the number of small mitochondria in the liver correlates negatively with glycogen granule saturation (Figure 14). In contrast, a negative correlation was found between the number of medium mitochondria and the intensity of hepatic lipid degeneration indicated by Sudan III staining, and a positive correlation was found with the saturation of glycogen granules in the liver (Figure 14). These correlations can be interpreted as indirect evidence of the functional insufficiency of small mitochondria. This is also supported by a positive correlation between the number of small mitochondria and mitochondrial LP products (Figure 14). In turn, LP products in mitochondria are generated by the incomplete reduction of molecular oxygen [68]. Moreover, there are reports that the intensity of mitochondrial LP is negatively correlated with cell lifespan [69].

It is likely that GCs could exert protective effects via their ability to enhance mitochondrial functions that provide cells with more energy during stress [70]. Overall, GC deficiency could mediate inflammatory responses in the liver, as GCs at low concentrations would be unable to downregulate the effects of proinflammatory cytokines on mitochondria. Of note, the blood Cort concentration correlated positively with the number of medium mitochondria and negatively with the number of small mitochondria in the current investigation (Figure 14). On the other hand, IL-1 correlated positively with the number of small mitochondria and negatively with the number of medium mitochondria (Figure 14). Currently, IL-1 is considered an inflammatory marker associated with lipid levels and the process of atherosclerosis [71].

It is well known that GCs block several inflammatory pathways. Low GCs concentrations in HA rats would have potentiated the effects of proinflammatory agents in the liver [72]. Information on the hepatic effects of GCs is contradictory. On the one hand, there is evidence of GC-induced fatty liver disease [73]. In the current study, signs of steatosis were observed in HA rats. On the other hand, some data support the hepatoprotective effect of GCs [74]. We propose that the hepatotoxic effects of GCs primarily characterize the side effects of steroid drugs. During stress, GCs limit the negative effects of proinflammatory cytokines on the liver so that these hormones have a hepatoprotective effect [75]. This view is supported by the negative correlation reported here between Cort and the intensity of hepatocytic lipid degeneration (Figure 14). Notably, a negative correlation between plasma Cort concentration and liver IL-1 and IL-2 concentrations was also evident (Figure 14). This correlation reflects the well-known antagonism between GCs and proinflammatory cytokines during stress. Overall, the decrease of plasma Cort along with the observed imbalances between proinflammatory and anti-inflammatory factors, appear to be highly involved in the pathogenesis of the stress-induced liver damage observed in HA rats. In the present study, the HA rats had elevated cholesterol and triglycerides. It is well known that patients with psychiatric disorders have a significantly increased risk of dying from cardiovascular disease. One mechanism linking anxiety and cardiovascular mortality is atherogenic dyslipidemia [76,77,78,79]. At the same time, positive correlations were found between the concentration of IL-1 in the liver with both plasma cholesterol and triglycerides (Figure 14). In turn, the signs of dyslipidemia in stressed rats were positively correlated with the intensity of lipid destruction. Previously, it was shown that rats with higher anxiety had increased low-density, proatherogenic lipoproteins, whereas the concentration of high-density, antiatherogenic lipoproteins was reduced, along with an increase in the atherogenicity coefficient [78]. Recently, we demonstrated a positive relationship between anxiety after PS and ECG disturbances [25]. Given that the aforementioned lipoprotein fractions are synthesized in the liver, a clear link between liver dysfunction and heart damage in HA rats seems evident.

We have previously demonstrated an association between hepatic dysfunction and the development of anxiety disorders [43]. The results presented here have expanded our understanding of the hepatic factors involved in the development of anxiety and its manifestations. For example, the AI correlated positively with the intensiveness of liver Sudan III stain and with blood cholesterol and triglyceride concentrations. In contrast, AI was correlated negatively with blood glucose. At the same time, there is a negative correlation between the glucose concentration in the blood and the concentration of IL-1 in the liver (Figure 14). In this context, it is appropriate to mention the property of proinflammatory cytokines to have a hypoglycemic effect [80]. However, low blood glucose may be the result of increased glucose consumption by the brain under conditions of neuroinflammation. The proinflammatory cytokine interleukin (IL)-1β upregulates GLUT1 in endothelial cells and astrocytes while it induces neuronal death in neuronal cell cultures [81]. It is likely that in highly anxious rats, elevated levels of proinflammatory cytokines create a background that contributes to the development of endothelial dysfunction and neuroinflammation [42,82,83]. In turn, during neuroinflammation, glucose consumption increases in brain regions such as the amygdala, which is actively involved in the development of anxiety-like behavior [51]. Moreover, increased glucose consumption in regions such as the amygdala may also occur in the presence of hypoglycemia [82].

Recently an interrelationship between hypoglycemia and neuroinflammation has been recognized [83]. Hypoglycemia was shown to induce neuroinflammation by (i) damage to the blood-brain barrier; (ii) cerebral edema formation; (iii) c-fos expression in the hippocampus; (iv) induction of oxidative stress; (v) increased TNF-α, IL-1β, and IL-6 mRNA in brain tissues.

The current findings suggest that strategies aimed at controlling excessive mitochondrial oxidative stress in PTSD may lessen liver damage in PTSD. For example, the antioxidant potential of resveratrol should lessen mitochondrial dysfunction. In fact, recently, Li et al. found that resveratrol decreased anxiety-like behavior in this rat model of PTSD [84,85,86]. Further investigations should examine the hepatoprotective effects of resveratrol in the PTSD rat model. Recently, our pilot studies showed the benefits of using resveratrol for this purpose [87]. Further studies should determine the ability of resveratrol and similar antioxidants and hepatoprotectors to correct or prevent liver mitochondrial dysfunction of hepatocytes in PTSD.

In summary, the findings of this investigation explain the development of liver dysfunction in HA rats. Numerous correlations between the size of mitochondria and/or markers of mitochondrial oxidative stress and hepatocellular damage, e.g., ALT and AST, indirectly confirm an orchestrating role of mitochondrial dysfunction in the origin and development of stress-induced liver damage. This point is also supported by correlations between liver histology and mitochondrial characteristics, especially indicators of mitochondrial oxidative stress. Moreover, the use of hepatoprotectors will reveal important details of the proposed mechanism for the development of liver damage through mitochondrial dysfunction.

## 4. Materials and Methods

In general, our methodological approach was based on the exposure of rats to stress capable of causing PTSD, identifying rats with high anxiety, and then analyzing molecular, biochemical, and structural characteristics of these rats that differed from rats with low anxiety. The stress was exposure to predator scent. High- and low-anxiety rats were identified by a behavioral test. The analytical procedures described below provide evidence that mitochondrial dysfunction is a probable mechanism of the hepatic damage we observed in rats with post-traumatic stress.

### 4.1. Experimental Animals

Male Wistar rats (215 ± 12 (SD)) g, n = 30) were placed 10 per cage and provided standard rat chow and water ad libitum. After acclimation, the rats were randomly divided into two groups: a control group (n = 10) and an experimental group (n = 20) that was exposed to cat urine scent for 15 min daily for 10 days. To allow the experimental PTSD to develop, the rats were then rested for 14 days under stress-free conditions [24]. During this time, one rat lost mass and was eliminated from the study. Control rats were housed under similar conditions for the same period under stress-free conditions. This experimental model of PTSD was developed by Cohen [14] and has been used in many subsequent studies of experimental PTSD [12,13]. The model is based on the well-known fear response of rodents to a predator and its smell. The immediate and delayed stress responses are characteristic of human responses to severe, traumatic stress.

The rats were sacrificed by an overdose of diethyl ether, decapitated, and blood was collected on experimental day 28. The mass of the rats and their liver was determined at necropsy. The liver was macroscopically assessed visually for size, color, and elasticity in response to gripping with tweezers. Relative mass (weight index) was calculated as the weight of the liver (WL) relative to rat body weight (BW), i.e., WL(g)/BW(g).

### 4.2. Behavioral Testing

The anxiety of all rats was evaluated with an elevated-plus-maze (EPM) test using the standard EPM apparatus TS0502-R3 (OpenScience, Moscow, Russia). The total duration of the test was 10 min. Control and experimental rats were tested together in a blind fashion. The behavior of rats in the EPM was recorded and tracked using a SMART video system and analyzed with SMART 3.0 software. The number of entries into the open and closed arms of the EPM and the time spent in the open and closed arms were recorded. Based on these measurements, the AI was calculated [17]: AI = 1 − {[(time in open arms/Σ time on maze) + (number of entries into open arms/Σ number of all entries)]/2}. An AI > 0.8 was considered a marker for the presence of high anxiety-like behavior; rats with AI > 0.8 were assigned to the HA group. Rats with AI < 0.8 were assigned to the LA group. The AI discriminant of 0.8 was set based on the AI distribution of naive rats measured in preliminary experiments for this study and on historical AIs of control rats, as reported earlier [25].

### 4.3. Blood and Tissue Collection and Storage

At necropsy, blood and the liver were collected. After centrifugation, blood plasma was stored in Eppendorf tubes at −70 °C. Liver tissue was stored in 10% buffered formalin for histological analyses and frozen in liquid nitrogen, and stored at −70 °C for biochemical analyses.

### 4.4. Measurement of Glucose Concentration and Lipidogram

Concentrations of glucose, total cholesterol (TC), high-density lipoprotein cholesterol (HDL-C), low-density lipoprotein cholesterol (LDL-C), and triglycerides were measured in blood with a CardioChek PA Analyzer (PTS Diagnostics, Whitestone, IN, USA).

### 4.5. ELISA

Plasma Cort was measured by ELISA (IBL, Hamburg, Germany). Plasma concentrations of IL-1, IL-2, IL-6, IL-10, IL-4 (Bender Medsystems, Wien, Austria), and Cort (IBL, Germany) were determined by ELISA. The microplate ELISA analyzer ANTHOS 2010 (Wien, Austria) was used to record the results.

### 4.6. Mitochondria Isolation

Mitochondria were isolated from liver tissue homogenates, according to Satav and Katyare [88].

### 4.7. Histomorphological and Histochemical Analysis

#### 4.7.1. Evaluation of Liver Lipids

For histological examination, liver tissue from the right lobe was processed for cryosectioning (Tissue-Tek Cryo3 Flex, Japan) to examine intra-hepatocytic lipids using Sudan III staining. Frozen sections with a thickness of 7 μm were stained with Sudan III (Sigma, St. Louis, MO, USA). For slides stained for lipids, glycogen, and protein, the optical density of section staining was measured on microphotographs with an Axioplan 2 Imaging microscope (Carl Zeiss, Berlin, Germany) using AxioVision (Carl Zeiss, Berlin, Germany) and ImageJ (NIH, Bethesda, MD, USA, version 3) software.

#### 4.7.2. Evaluation of Glycogen Deposition in Liver

Pieces of rat liver, ~4 mm^3^, were fixed in 10% neutral formaldehyde for at least 48 h at 20–22 °C and then embedded in paraffin blocks. The samples were dehydrated with alcohol, cleared with xylene, and embedded in melted paraffin. Histological sections approximately 6 µm thick were made. The tissue sections were deparaffinized in xylene and rehydrated through graded ethanol. Histochemical staining was performed using standard techniques to reflect glycogen deposition in the liver [89]. Sections were oxidized with periodic acid for 10 min and stained with Schiff reagent for 15 min at room temperature. Tissue sections digested with amylase were used as negative controls. This periodic acid Schiff technique is widely used to detect glycogen in tissues [90].

#### 4.7.3. Evaluation of Total Protein in Liver

Bromophenol blue (BRB) dye is widely used as a laboratory indicator to detect total proteins. The tissue sections were deparaffinized in xylene and rehydrated through graded ethanol. Sections were stained for 2 h at room temperature in a mixture of 2% acetic acid solution with 1% sublimate solution and 0.05% BRB solution and then washed for 5 min in 0.5% acetic acid. Sections stained and washed were dehydrated in alcohol (70°, 96°) to avoid turbidity, clarified in carbol xylene, and then placed in a synthetic medium and covered with a coverslip.

### 4.8. Transmission Electron Microscopy (TEM)

Liver samples 2 mm^3^ were fixed with a 2.5% glutaraldehyde solution in phosphate buffer (pH 7.4). They were moved in a 1% solution of osmium tetrachloride (OsO_4_) and dehydrated in ethanol according to standard methods. During dehydration, 1% uranyl acetate was contrasted with 70% ethanol and poured into the epon-araldite mixture according to the standard method [91]. Semi-thin sections were stained with a 25% toluidine blue solution (Sigma, St. Louis, MO, USA) to select optimal sections. Ultrathin sections of 60–90 nm were prepared using the LKB-III ultratome (LKB Produkter, Uppsala, Sweden), stained with uranyl acetate and lead citrate, and examined in a transmission electron microscope JEM-100CX (JEOL, Tokyo, Japan). Photographs of the preparations were taken using the Erlangshen camera Gatan ES500W (model 782, Gatan Inc., Pleasanton, CA, USA).

### 4.9. Measurement of Superoxide Dismutase (SOD) Activity in Liver

The assay for determination of SOD-activity is based on the ability of SOD to compete with nitroblue tetrazolium (NBT) for superoxide anions formed as a result of the aerobic interaction of NADH and phenazine methosulfate. The content of NBT reduction products (nitroformazan) was performed at a wavelength of 540 nm [92].

### 4.10. Content of LP Products in Liver and in Hepatic Mitochondria

The tissue content of lipid peroxidation products was assayed by an extraction spectrophotometric method [93]. This method allows differential measurement of acyl peroxides among phospholipids extracted from the propanol-2 phase, along with non-esterified intermediates of fatty acid peroxides extracted from the heptane phase. Results were expressed as oxidation indices E278/220 for relative contents of ketodienes and conjugated trienes.

### 4.11. Data Analyses

Data were analyzed using the programs SPSS 24 (IBM, New York, NY, USA), STATISTICA 10.0 (StatSoft, Tulsa, OK, USA), RStudio (RStudio, Boston, MA, USA), and Excel (Microsoft, Redmond, WA, USA). The normality of data distributions was tested using the Shapiro–Wilk procedure. Data are presented as mean ± SEM or median (25th–75th percentile). Normally distributed data were analyzed with a parametric, one-factor ANOVA followed by Tukey’s post hoc tests to compare all outcome measures between respective groups. Non-normally distributed data were analyzed using a nonparametric, one-factor Kruskal–Wallis ANOVA, followed by Dunn’s tests for pairwise comparisons between respective groups. Relationships between all measured variables were examined using Spearman correlation analysis. A correlation matrix illustrates these results. For all procedures, *p* < 0.05 was considered statistically significant.

## 5. Conclusions

The main finding of this study is the association between liver mitochondrial dysfunction, liver damage, and the presence of anxiety disorders in PTSD. This relationship reflects the importance of hepatic disorders in the pathogenesis of PTSD. Experimental PTSD induced significant liver damage in HA rats, whereas rats that were resistant to stress-induced anxiety were also protected from liver damage. The liver damage in the HA phenotype was associated with oxidative stress and with mitochondrial dysfunction. The mitochondrial dysfunction was characterized by an increase in the number of small mitochondria suggesting compromised energy metabolism. The hepatic dysfunction in the HA rats was expressed in the increase of plasma triglycerides and cholesterol with a simultaneous decrease of plasma glucose. In turn, a decrease in hepatic glucose predisposes the development of neuronal dysfunction, resulting in an increase in anxiety. The validity of these assumptions is confirmed by numerous relevant correlations. The data are schematically summarized in a putative scheme that explains interrelationships among neurohepatological mechanisms of anxiety development. The current finding should stimulate further studies that will reveal important details related to the identification at the molecular level of pathogenetic mechanisms in the development of hepatic mitochondrial dysfunction.

## Figures and Tables

**Figure 1 ijms-24-13012-f001:**
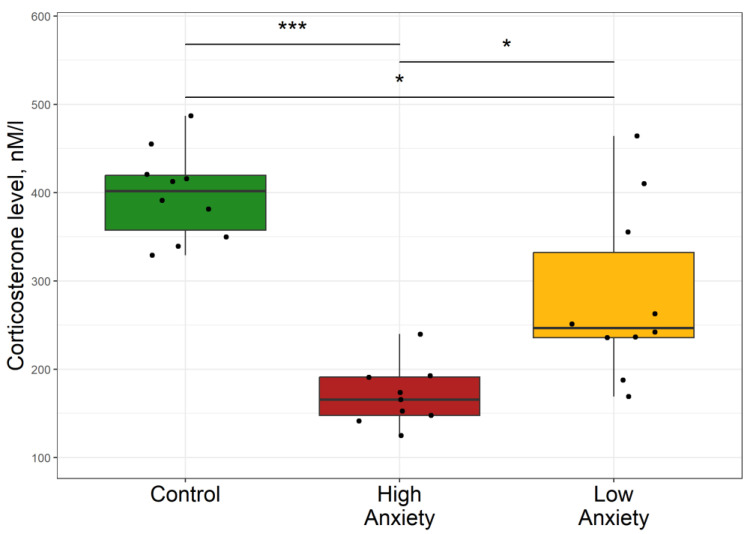
Differences in plasma corticosterone concentrations between the control, low anxiety, and high anxiety groups of rats. In this and in similar figures, the boxes include the middle 50% of the data, i.e., from the 25th to the 75th percentile, with the median value shown by the horizontal line. The whiskers include data that fall within 1.5 times the interquartile range. *p*-values were determined by non-parametric analysis. Control n = 10; High Anxiety n = 9; Low Anxiety n = 10. * *p* < 0.05, *** *p* < 0.001.

**Figure 2 ijms-24-13012-f002:**
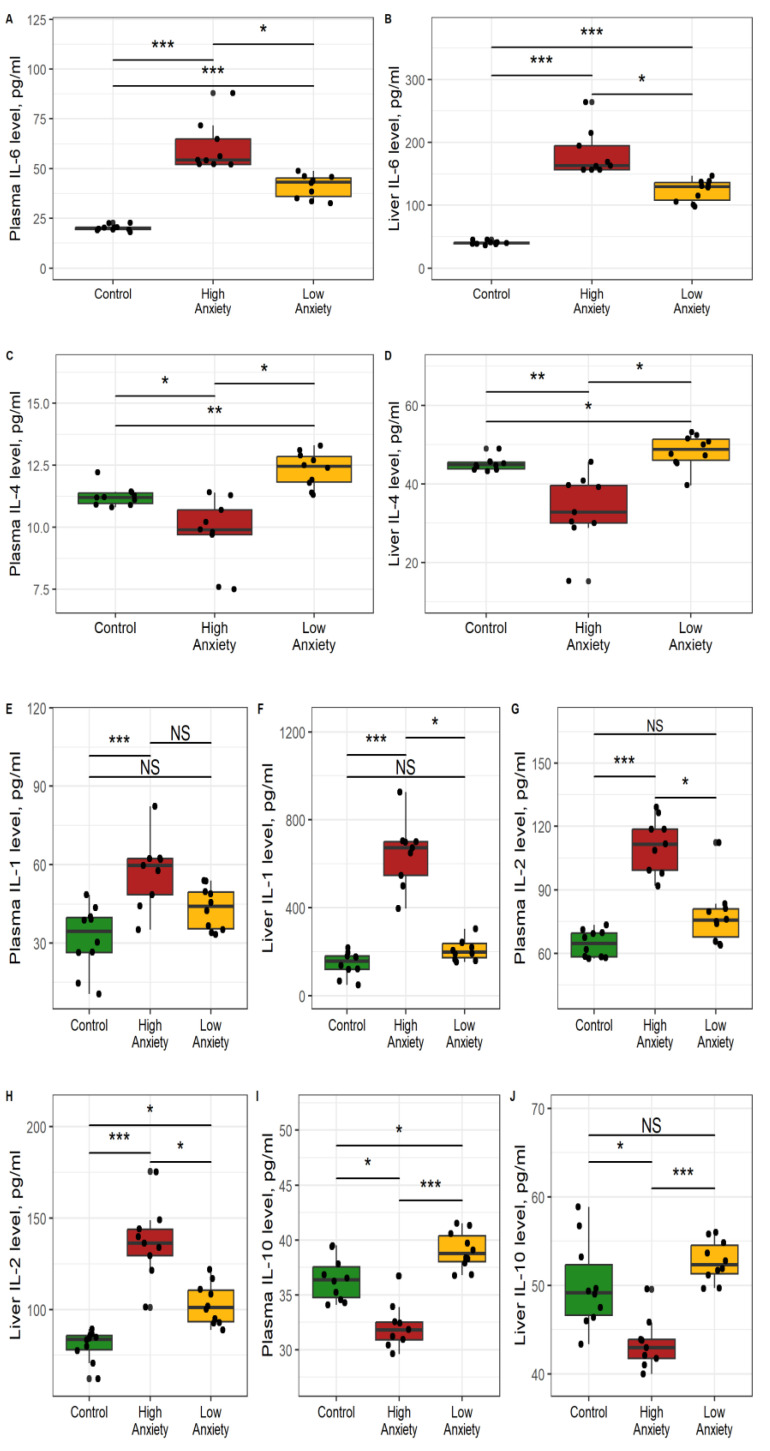
Effects of PS on cytokine concentrations in plasma and liver. * *p* < 0.05, ** *p* < 0.01, *** *p* < 0.001, NS, not significant. (**A**) plasma IL-6; (**B**) liver IL-6; (**C**) plasma IL-4; (**D**) liver IL-4; (**E**) plasma IL-1; (**F**) liver, IL-1; (**G**) plasma IL-2; (**H**) liver IL-2; (**I**) plasma IL-10; (**J**) liver IL-10.

**Figure 3 ijms-24-13012-f003:**
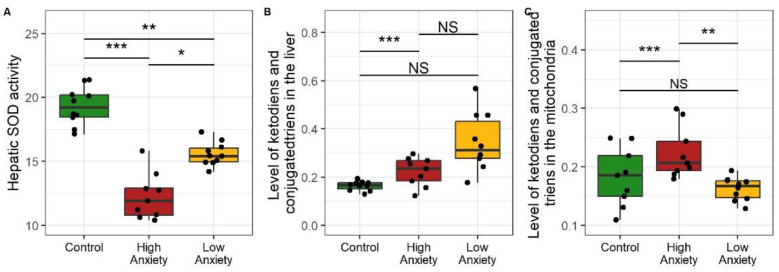
Oxidative stress in the liver as reflected by hepatic superoxide dismutase (SOD), ketodienes, and conjugated trienes. * *p* < 0.05, ** *p* < 0.01, *** *p* < 0.001, NS, not significant. (**A**) Hepatic SOD activity; (**B**) Hepatic concentrations of ketodienes and conjugated trienes; (**C**) Mitochondrial concntrations of ketodienes and conjugated trienes. SOD activity is expressed in units min^−1^ mg protein; ketodienes and conjugated trienes are expressed as oxidation indices E278/220.

**Figure 4 ijms-24-13012-f004:**
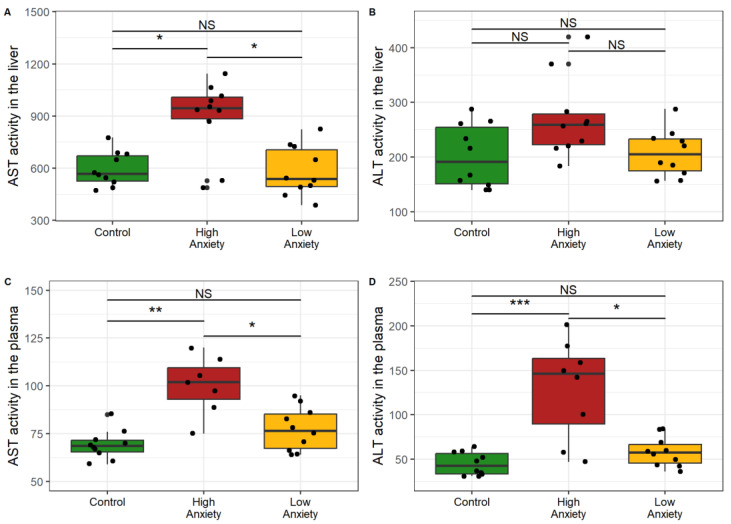
AST and ALT activity in plasma and liver. * *p* < 0.05, ** *p* < 0.01, *** *p* < 0.001, NS, not significant. (**A**) AST activity in liver; (**B**) ALT activity in liver; (**C**) AST activity in plasma; (**D**) ALT activity in plasma.

**Figure 5 ijms-24-13012-f005:**
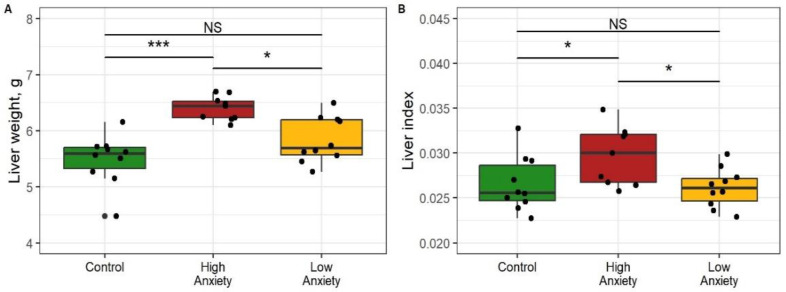
Liver weight and weight index values in control and stress-exposed rats. * *p* < 0.05, *** *p* < 0.001, NS, not significant. (**A**) Liver weight; (**B**) Liver index.

**Figure 6 ijms-24-13012-f006:**
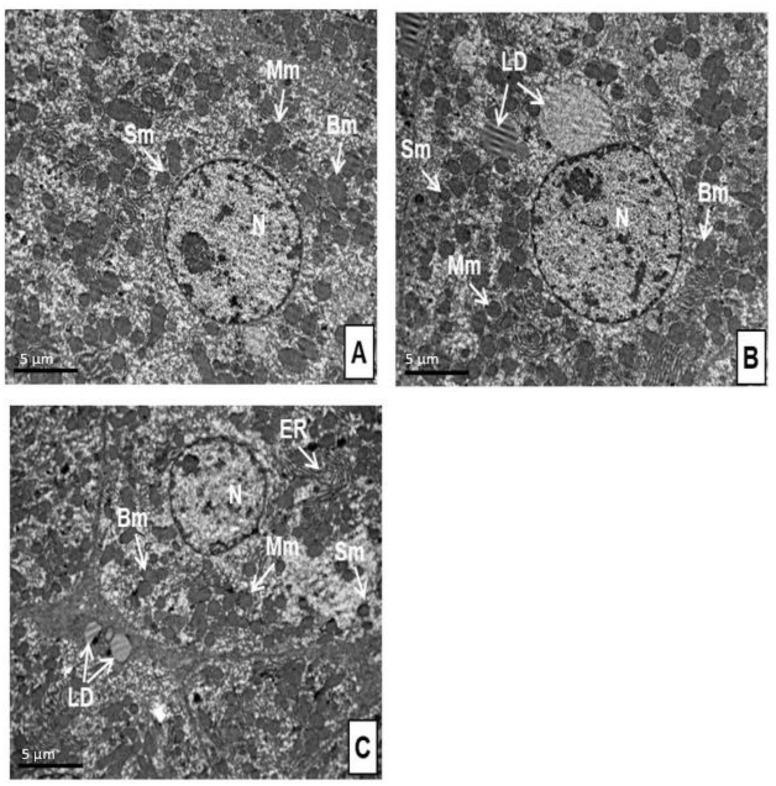
Ultrastructure of liver cells of control rats (**A**), LA rats (**B**), and HA (**C**) rats. N, nucleus; Sm, small mitochondria; Mm, medium mitochondria; Bm, big mitochondria; ER, endoplasmic reticulum; LD, lipid droplets. Bar = 5 µm.

**Figure 7 ijms-24-13012-f007:**
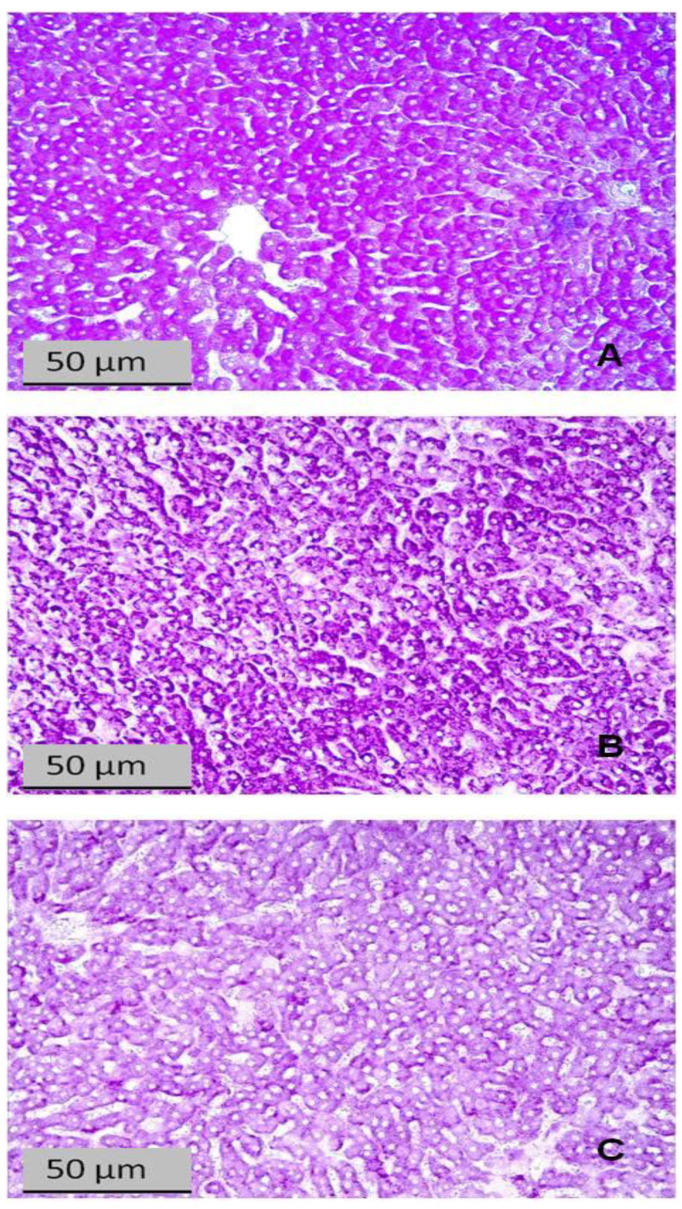
Liver sections stained with Schiff reagent for glycogen (PAS reaction). (**A**) Control rat; (**B**) LA rat; (**C**) HA rat.

**Figure 8 ijms-24-13012-f008:**
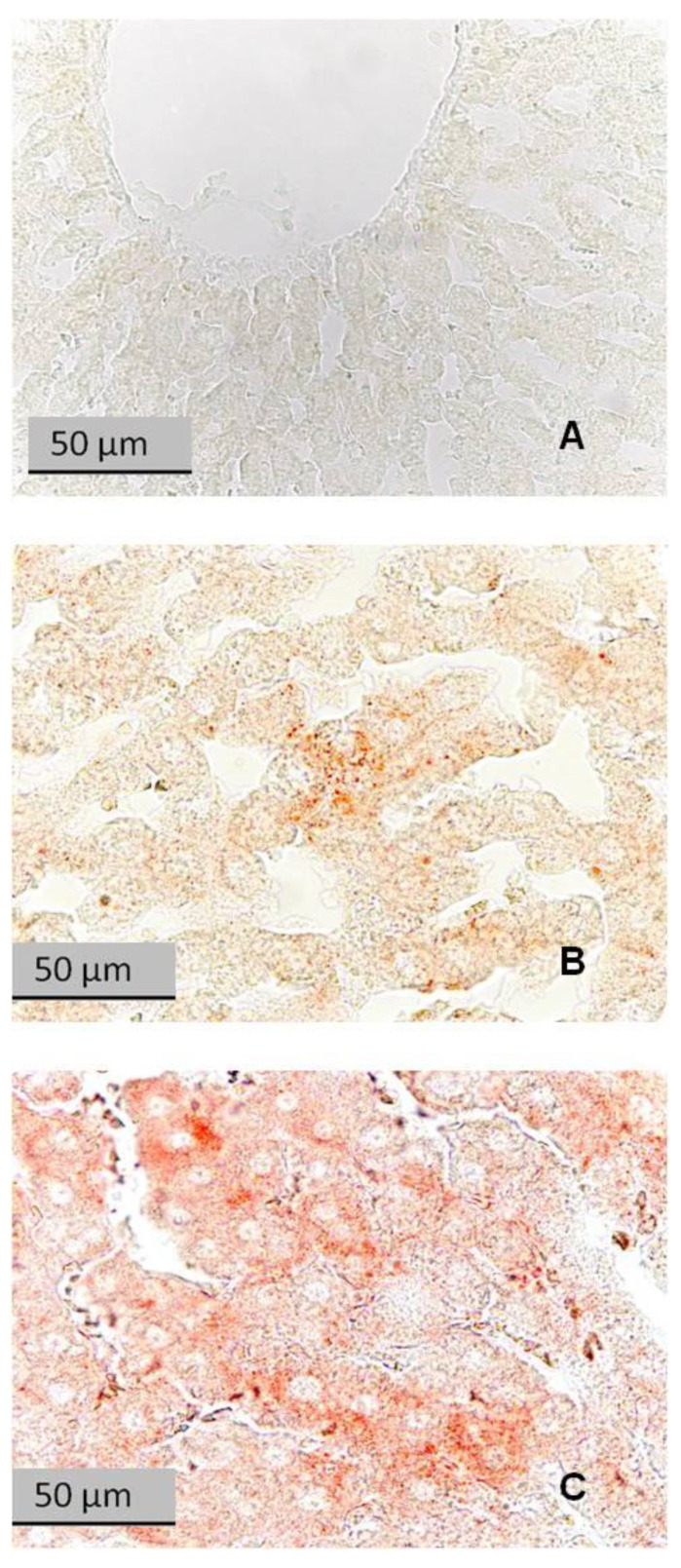
Liver sections stained with Sudan III for lipids. (**A**) Control rat; (**B**) LA rat; (**C**) HA rat. Reddish-orange spots in hepatocytes are neutral lipids.

**Figure 9 ijms-24-13012-f009:**
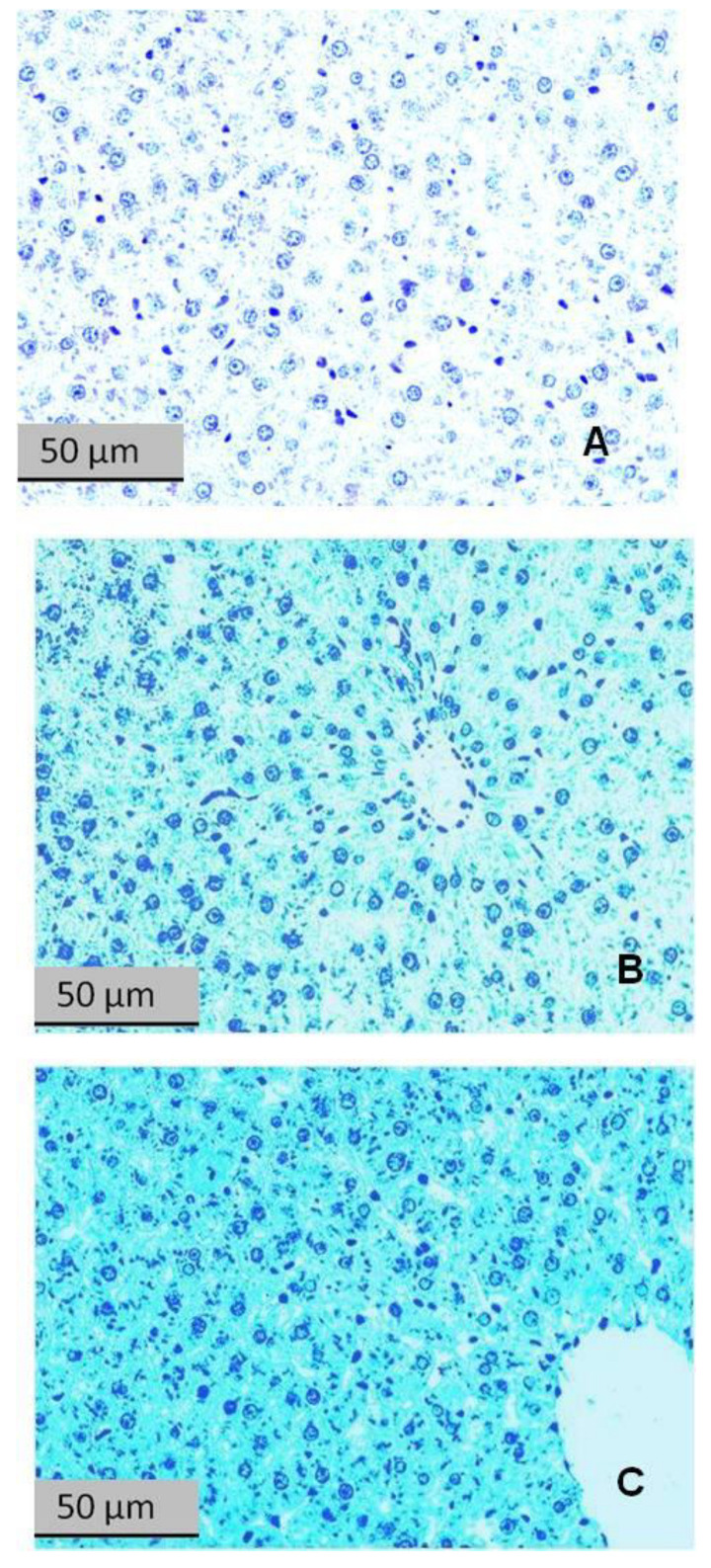
Liver sections stained with bromophenol blue for protein content. (**A**) Control rat; (**B**) LA rat; (**C**) HA rat.

**Figure 10 ijms-24-13012-f010:**
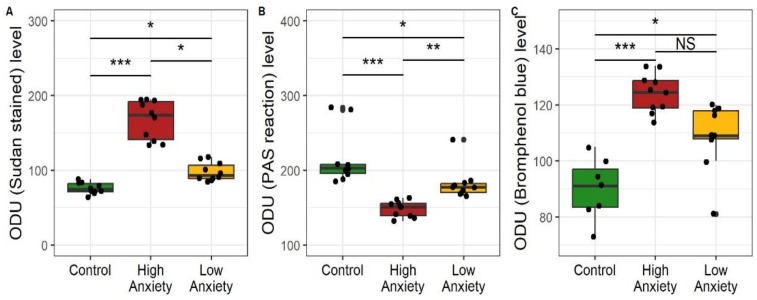
Optical density (optical density units, ODU) of liver sections stained for (**A**) neutral fats, Sudan III, (**B**) glycogen, periodic acid-Schiff (PAS) reaction, (**C**) protein, bromophenol blue. * *p* < 0.05, ** *p* < 0.01, *** *p* < 0.001, NS, not significant.

**Figure 11 ijms-24-13012-f011:**
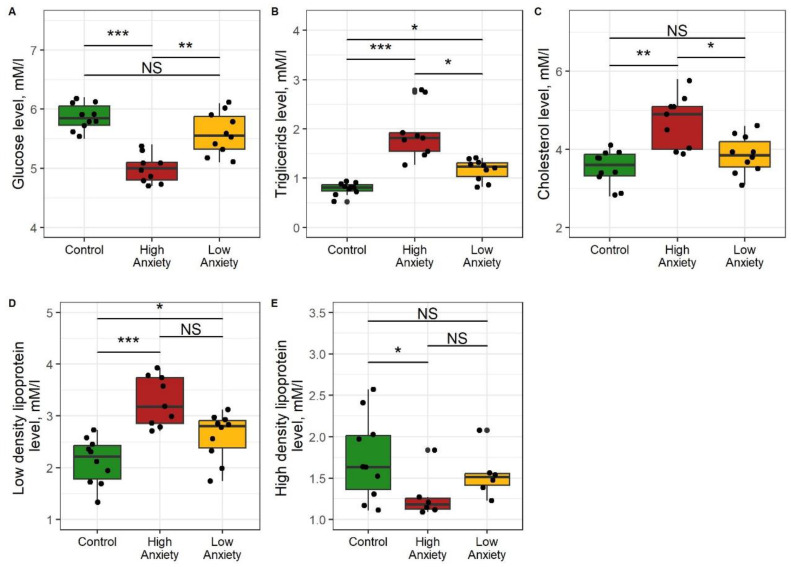
Glucose (**A**), triglycerides (**B**), cholesterol (**C**), and low-density (**D**) and high-density (**E**) lipoprotein concentrations in blood of control, HA, and LA rats. * *p* < 0.05, ** *p* < 0.01, *** *p* < 0.001, NS, not significant.

**Figure 12 ijms-24-13012-f012:**
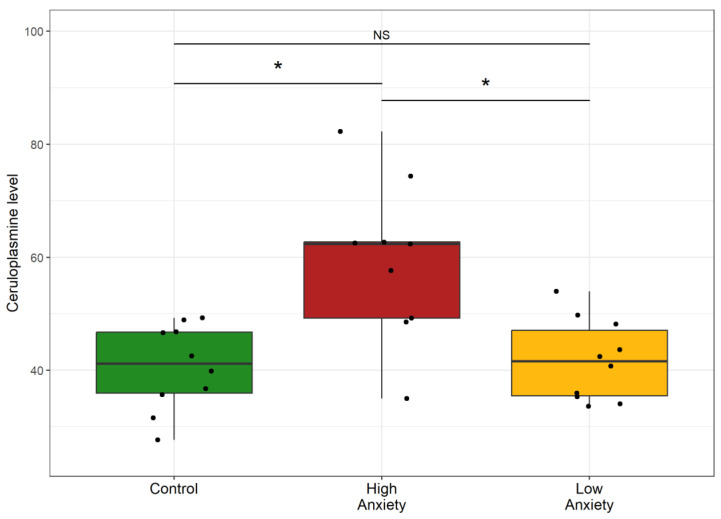
Ceruloplasmin in control and stress-exposed rats. * *p* < 0.05, NS, not significant.

**Figure 13 ijms-24-13012-f013:**
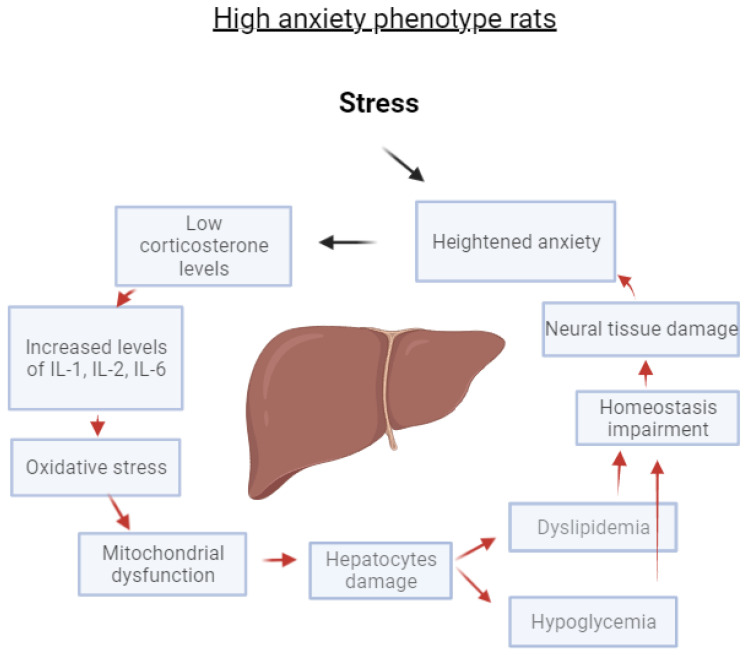
A positive feedback loop explains the interrelations between liver damage and anxiety in HA rats exposed to experimental PS.

**Figure 14 ijms-24-13012-f014:**
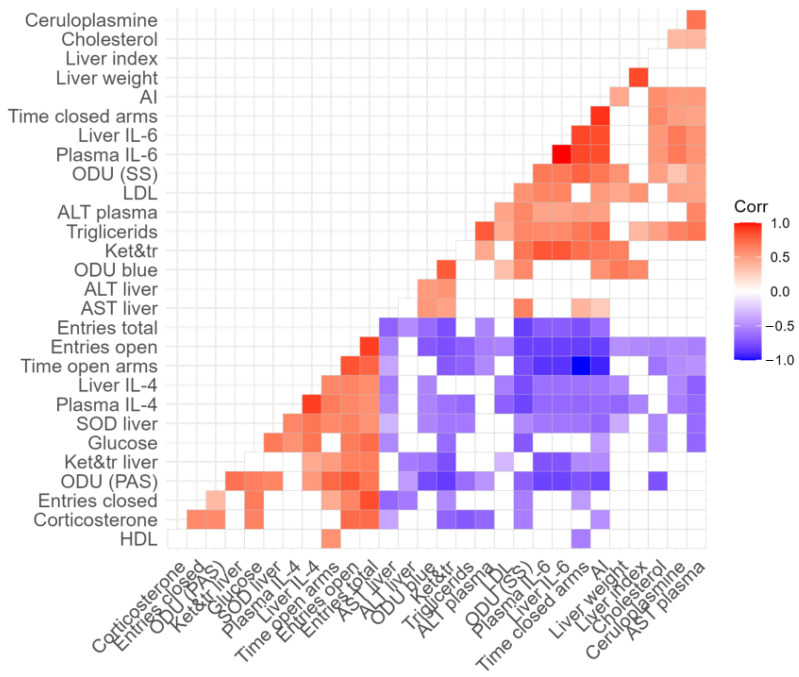
Correlation matrix showing the relationship between measured variables in PS-exposed rats. Respective r values are indicated by the color bar below the matrix. *p* < 0.05 for all colored correlations. Details are shown in the Appendix A.

**Table 1 ijms-24-13012-t001:** Results of EPM behavioral experiments.

Variable	Control Rats (n = 10)	HA Rats (n = 9)	LA Rats (n = 10)
Time spent in open arms (s)	255 (242.5; 270)	10 (5; 10) *^,^#	140 (122.5; 193.8)
Time spent in closed arms (s)	345 (330; 357.5)	590 (590; 595) *^,^#	460 (406.2; 477.5)
Number of entries into open arms	6 (5.25; 6.75)	1 (1; 1) *^,^#	5 (5; 6)
Number of entries into closed arms	10 (9; 10)	6 (3; 9) *	9 (8; 10.75)
Anxiety index (AI)	0.60 (0.58; 0.61)	0.93 (0.9; 0.94) *^,^#	0.68 (0.64; 0.73)

Data are median (first quartile; third quartile). * Significantly different from control rats. See text for specific *p* values. # Significantly different from LA rats.

**Table 2 ijms-24-13012-t002:** Morphometric data of rat mitochondria.

Variable	Mitochondria Subtypes	Group		
		Control Rats (n = 10)	LA Rats (n = 10)	HA Rats (n = 9)
Mitochondrial area (μm^2^)	Small mitochondria	3.22 (3.6; 3.67)	3.57 (3.46; 4.12)	2.07 (1.89; 2.39) **^,^##
	Medium mitochondria	13.98 (13; 15)	7.26 (7.22; 7.87) *	6.45 (5.9; 6.85) ***^,^#
	Large mitochondria	31.14 (27.77; 35.96)	15.78 (14.11; 16.82) ***	10.78 (9.89;11.86) ***^,^##
Number of mitochondria per 100	Small mitochondria	24 (21.25; 25.75)	52 (49.5; 53.75) *	91 (88.25; 92) ***^,^#
	Medium mitochondria	34.5 (33; 37.75)	82.5 (79.5; 85.5) ***	58.5 (57.25; 61) *^,^#
	Large mitochondria	10 (9.25; 11.75)	9 (8; 10)	9.5 (8; 10)
	Total number	67 (65.25; 74)	145 (142; 148) **	158.5 (153.5; 163) ***^,^#

Data are median (first quartile; third quartile) * *p* < 0.05, ** *p* < 0.01, *** *p* < 0.001 vs. control rats; # *p* < 0.05, ## *p* < 0.001 HA rats vs. LA rats.

## Data Availability

The data presented in this study are available in article.

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
