# Peer review of "Unveiling the Link: Exploring Mitochondrial Dysfunction as a Probable Mechanism of Hepatic Damage in Post-Traumatic Stress Syndrome"

_ijms, 2023, doi:10.3390/ijms241613012_

Round 1
Reviewer 1 Report (New Reviewer)
Comments and Suggestions for Authors
The authors of this paper used a chronic predator stress model to analyze how post-traumatic stress disorder (PTSD) affects liver function. They reported a wide array of liver damage phenotypes including accumulation of lipid droplets, increase in proinflammatory cytokines and mitochondrial concentrations of liver peroxidation products, and changes in mitochondria numbers and morphology. In the discussion, the authors proposed a model in which liver damage induced hypoglycemia induces neural tissue damage to increase anxiety. There are two major weaknesses: 1. The title of this paper claims “focus on mitochondrial dysfunction”, however, the only data on mitochondria is that in HA rats, the number of small mitochondria significantly increased. The mechanism of the increase was not addressed. The causal relationship between the change in mitochondria and the increased oxidative stress was not addressed. 2. The entire study is purely observational in nature. All the results the authors obtained are consistent with their hypothesis but do not establish causal relationship. For instance, if the authors want to argue that liver damage induced hypoglycemia induces neural tissue damage, they should provide data or cite references to show A. artificial induction of hypoglycemia induces neural tissue damage; B. rescue of hypoglycemia in PS model reduces neural tissue damage. Without these types of experiments, all the results could be coincidental. Two small issues: 1. The asterisks in figures 1 and 2 should be closer to the straight lines drawn between groups so that they are less confusing; 2. PS induced IL-10 increase in plasma but decrease in liver, this contradictory result should be discussed.
Author Response
Please see the attachment

Reviewer 2 Report (New Reviewer)
Comments and Suggestions for Authors
13 June 2023
The decision on the manuscript, titled “Mechanisms of hepatic damage in PTSD: Focus on mitochondrial dysfunction” by Kondashevskaya MV et al., submitted to International Journal of Molecular Sciences (IJMS)
Manuscript ID: ijms-2471328
Dear Authors,
Developing efficient interventions that can lessen the effect of mitochondrial dysfunction on the liver in people with post-traumatic stress disorder (PTSD) is the current challenge. Kondashevskaya and colleagues, in the present research article entitled ‘Mechanisms of hepatic damage in PTSD: Focus on mitochondrial dysfunction’, investigated the effects of chronic predator stress (PS) on the liver and its influence on anxiety-related behavior, cytokine levels, oxidative stress, and liver enzyme activity in rats. Here, the authors suggested that PS leads to the development of anxiety phenotypes, with high-anxiety (HA) rats exhibiting greater anxiety levels compared to low-anxiety (LA) and control rats. HA rats also showed significant alterations in liver function, including increased levels of proinflammatory cytokines, oxidative stress, and liver enzyme activity. The findings suggested a close interaction between the liver and the brain in response to stress, highlighting the liver-brain axis as a potential target for studying stress-related disorders such as PTSD and liver dysfunction.
The main advantage of this manuscript is that it addresses a pertinent and interesting issue while also providing a comprehensive analysis of the mechanisms of hepatic damage in PTSD, with a particular focus on the role of mitochondrial dysfunction and its potential implications for the development of new treatments and interventions.
In general, I think the idea of this article is really interesting, and the authors’ fascinating observations on this timely topic may be of interest to the readers of the IJMS. However, some comments, as well as some crucial evidence that should be included to support the author’s argumentation, needed to be addressed to improve the quality of the manuscript, its adequacy, and its readability prior to its publication in the present form.
Please consider the following comments:
1. Title: This is the most important section of the manuscript. Please present a concise and self-explanatory title stating the most important findings of this study. Suggestions: "Unveiling the Link: Exploring Mitochondrial Dysfunction as a Mechanism of Hepatic Damage in Post-Traumatic Stress Syndrome” [1-3]. Please avoid using abbreviations in the title.
2. A graphical abstract that will visually summarize the main findings of the manuscript is highly recommended.
3. Abstract: I would like the authors to make as much effort for this section as for the rest of the manuscript. Please present the abstract with 200 words according to the guidelines of the journal [4], focusing on proportionally presenting the background, results, methods, discussion, and conclusion (without the headings of subsections). The background should include the general background (one to two sentences), the specific background (two to three sentences), and "the current issue addressed to this study" (one sentence), leading to the objectives. In this subsection, I would like the authors to lay out basic information, a problem statement, and their motivation to break off. The methods should clarify the authors’ approach, such as study design and variables, to solve the problem and/or make progress on the problem. The conclusion should open with one sentence describing the main result using such words like “Here we show”, which should be followed by statements such as the potential and the advance this study has provided in the field and finally “a broader perspective (two to three sentences) readily comprehensible to a scientist in any discipline” [5-7].
4. Keywords: Please list ten keywords chosen from Medical Subject Headings (MeSH) [8] (and use as many as possible in the title and in the first two sentences of the abstract [7,9].
5. In general, I advise authors to cite more sources to support their assertions, particularly in the introduction of this research article, which I consider to be deficient. Thus, I recommend the authors attempt to expand the topic of their article, as the bibliography is too concise. Nevertheless, I believe that less than 60 or 70 articles are too few for a research article. Therefore, I suggest the authors focus their efforts on researching relevant literature; in my opinion, adding more citations will help provide a better and more accurate background to this study.
6. The manuscript would benefit from a clear and concise statement of the research objectives or hypotheses at the beginning of the introduction, that would help guide readers and provide a focus for the study.
7. Introduction: The authors need to fully reorganize this section with “several paragraphs” made up of “about 1000 words”, introducing information on the main constructs of this protocol, which should be understood to a reader in any discipline and make persuasive enough to put forward the main purpose of current research the author has conducted and the specific purpose the author has intended by this protocol. I would like to encourage the authors to present the introduction starting with the general background, proceeding to the specific background, rationales, and finally the current issue addressed to this study, leading to the objectives. Those main structures should be organized in a logical and cohesive manner [10].
8. In this regard, I believe that this section would benefit from more context and background information on pathophysiological mechanisms, which explain how chronic predator stress can lead to alterations in liver function and anxiety-related behavior. In this regard, to enhance the understanding of the pathophysiology, here the Authors should include information on the known physiological responses to stress, such as the activation of the hypothalamic-pituitary-adrenal (HPA) axis and the release of stress hormones like cortisol (DOI: 10.17219/acem/165944; DOI: 10.3390/ijms24065926; https://doi.org/10.3390/biomedicines11051465). Additionally, they could explore the potential role of inflammation, oxidative stress, and immune dysregulation in the context of chronic stress and liver dysfunction to provide a more comprehensive understanding of the pathophysiological processes involved (https://doi.org/10.1016/j.neubiorev.2023.105163; DOI: 10.3390/biomedicines11030945; https://doi.org/10.3389/fpsyt.2022.927075). In addition, the following works may enhance the value of this manuscript, including but not limited to: https://doi.org/10.3390/healthcare10050818; https://doi.org/10.3390/ijms23094881; https://doi.org/10.3390/brainsci12030387; https://doi.org/10.3390/ijms21175986; https://doi.org/10.3390/cells11162607; https://doi.org/10.3390/biomedicines9101293.
9. Effects of predator stress on plasma and hepatic cytokines: I would suggest rewriting this section more accurately. In my opinion, although the results regarding the cytokine concentrations are presented, it would be beneficial to provide more interpretation or discussion of these findings. What do the differences in cytokine levels suggest about the impact of predator stress on the immune response in the liver? Are these changes consistent with previous studies, or do they provide novel insights? I suggest closing results section with a paragraph which puts the results into a more general context. Also, please present figures in color.
10. Discussion: I would like the authors to present this section by opening with an introductory paragraph, followed by a summary of the previous section. Then, I expect the authors to develop arguments clarifying the potential of this study as an extension of the previous work, the implication of the findings, how this study could facilitate future research, the ultimate goal, the challenge, the knowledge and technology necessary to achieve this goal, the statement about this field in general, and finally the importance of this line of research. It is particularly important to present its limits, merits, and potential translation into clinical practice [11,12].
11. Material and Methods: I recommend opening this section with a short introductory paragraph and citing more references to ensure the reliability and integrity of the evidence in the study design the authors built and the methodology they have decided to apply. I also believe that this section would benefit from a clearer structure and better organization of the flow of information. For example, I would suggest that you clearly describe the experimental design and outline the steps taken to conduct the study, as well as provide more information about the type of stressor used (e.g., predator stress, social stress), the intensity or duration of the stressor, and the frequency of exposure. If applicable, authors should also mention any control groups or sham stress procedures used to differentiate the effects of stress from other factors and explain the specific behavioral assessments or tests used to evaluate anxiety-related behavior.
12. In my opinion, the ‘Conclusions’ paragraph would benefit from some thoughtful as well as in-depth considerations by the authors, because as it stands, it lists down all the main findings of the research, without really stressing the theoretical significance of the study. Authors should make an effort, trying to explain the theoretical implication as well as the translational application of their research.
13. Tables and Figures: According to the journal’s guidelines, please provide a short explanatory caption for the table within the text.
14. References: Authors should consider revising the bibliography, as there are several incorrect citations. Indeed, according to the journal’s guidelines [4], Please use et al. after ten names of authors.
Overall, the manuscript contains 14 figures, no tables, and 57 references. I believe that the manuscript may have merits in presenting its rigorous scientific approach, its clear and concise presentation of complex information, and its potential to advance our understanding of the relationship between mental health and physical health. I hope that, after careful revisions, the manuscript can meet the journal’s high standards for publication. I declare no conflict of interest regarding this manuscript.
Best regards,
Academic Editor
References:
1. https://plos.org/resource/how-to-write-a-great-title/
2. https://www.nature.com/nature-index/news-blog/how-to-write-a-good-research-science-academic-paper-title
3. https://www.indeed.com/career-advice/career-development/catchy-title
4. https://www.mdpi.com/journal/ijms/instructions
5. https://www.scribbr.com/dissertation/abstract/
6. https://writing.wisc.edu/handbook/assignments/writing-an-abstract-for-your-research-paper/
7. https://www.ncbi.nlm.nih.gov/pmc/articles/PMC7144240/
8. https://meshb.nlm.nih.gov/
9. https://pubmed.ncbi.nlm.nih.gov/30930712/
10. https://dept.writing.wisc.edu/wac/writing-an-introduction-for-a-scientific-paper/
11. https://doi.org/10.3163/1536-5050.103.2.001
12. https://www.scribbr.com/dissertation/discussion/
Comments on the Quality of English Language
13 June 2023
The decision on the manuscript, titled “Mechanisms of hepatic damage in PTSD: Focus on mitochondrial dysfunction” by Kondashevskaya MV et al., submitted to International Journal of Molecular Sciences (IJMS)
Manuscript ID: ijms-2471328
Dear Authors,
Based on the English proficiency assessment, it is noted that minor editing of the English language is required. While the overall communication is clear and understandable, there are some areas that could benefit from slight improvements in grammar, syntax, and word choice. Attention to detail, such as refining sentence structure and ensuring proper tense usage, will enhance the overall coherence and fluency of the written work. With some minor editing adjustments, the English language proficiency can be further enhanced.
Best regards,
Reviewer
Round 2
Reviewer 1 Report (New Reviewer)
Comments and Suggestions for Authors
The authors have successfully addressed the major concerns of the reviewers.
Author Response
Dear Reviewer:
Thank you very much for the favorable comments about our research. Your comments were very useful and motivated us to improve the text.
Reviewer 2 Report (New Reviewer)
Comments and Suggestions for Authors
30 July 2023
The 2nd review on the manuscript, titled “Mechanisms of hepatic damage in PTSD: Focus on mitochondrial dysfunction” by Kondashevskaya MV et al., submitted to International Journal of Molecular Sciences (IJMS)
Manuscript ID: ijms-2471328
Dear Authors,
In the present research article, entitled “Mechanisms of hepatic damage in PTSD: Focus on mitochondrial dysfunction,” Kondashevskaya and colleagues, investigated the effects of chronic predator stress on the liver and its influence on anxiety-related behavior, cytokine levels, oxidative stress, and liver enzyme activity in rats. I am pleased to see that the authors have attempted to revise the manuscript in the peer review session. Nevertheless, the revisions remain partial in regard to my review report. Prior to publication, I respectfully request that the authors consider my comments and revise the manuscript to meet the high standards of the journal.
Please consider the following comments:
1. Abstract: Please ensure that each section is presented in a proportional amount of detail without using subheadings. The conclusion is unproportionally short: I would like the authors to make as much effort for this section as for the rest of the manuscript. Please present the abstract with 200 words according to the guidelines of the journal [1], focusing on proportionally presenting the background, results, methods, discussion, and conclusion (without the headings of subsections). The background should include the general background (one to two sentences), the specific background (two to three sentences), and the current issue addressed to this study (one sentence), leading to the objectives. In this subsection, I would like the authors to lay out basic information, a problem statement, and their motivation to break off. The methods should clarify the authors’ approach, such as study design and variables, to solve the problem and/or make progress on the problem. The conclusion should open with one sentence describing the main result using such words like “Here we show”, which should be followed by statements such as the potential and the advance this study has provided in the field and finally a broader perspective (two to three sentences) readily comprehensible to a scientist in any discipline [2–4].
2. Keywords: Please verify that the listed keywords are included in the MeSH index: Please list ten keywords chosen from Medical Subject Headings (MeSH) [5].
3. Results: I recommend removing all statistical values in the text and present them in the tables. I suggest presenting figures in colors and closing this section with a paragraph that puts the results into a more general context.
4. Material and Methods: I recommend opening this section with a short introductory paragraph.
5. Tables and Figures: According to the journal’s guidelines, please provide a short explanatory caption for the table within the text.
6. References: Please follow the journal’s guidelines [4]. Please list authors’ name with commas and semicolons and use et al. after ten names of authors; abbreviated journal names end with periods; and volume numbers be italicized.
Overall, the manuscript contains 14 figures, two tables, and 93 references. I believe that the manuscript may have merits in presenting its rigorous scientific approach, its clear and concise presentation of complex information, and its potential to advance our understanding of the relationship between mental health and physical health. I hope that, after careful revisions, the manuscript can meet the journal’s high standards for publication. I declare no conflict of interest regarding this manuscript.
Best regards,
Academic Editor
References:
1. https://www.mdpi.com/journal/ijms/instructions
2. https://www.scribbr.com/dissertation/abstract/
3. https://writing.wisc.edu/handbook/assignments/writing-an-abstract-for-your-research-paper/
4. https://www.ncbi.nlm.nih.gov/pmc/articles/PMC7144240/
5. https://meshb.nlm.nih.gov/
Comments on the Quality of English Language
30 July 2023
The 2nd review on the manuscript, titled “Mechanisms of hepatic damage in PTSD: Focus on mitochondrial dysfunction” by Kondashevskaya MV et al., submitted to International Journal of Molecular Sciences (IJMS)
Manuscript ID: ijms-2471328
Dear Authors,
After evaluating the document, it is clear that some minor revisions are necessary for the English language. The document contains grammatical errors. Moreover, certain sentences are unclear and require rephrasing to enhance readability. Although the content of the document is informative and well-organized, the quality of the English language needs improvement to ensure clarity and conciseness. Therefore, it is essential to perform minor editing to enhance the overall quality and readability of the document.
Best regards,
Reviewer
Author Response
Dear Academic Editor
Thank you very much for the favorable comments about our research and for your valuable comments and detailed suggestions for improving the entire text. We regret that not all of your wishes were fulfilled in the first round of review. Now we have tried to completely correct the text.
- Abstract: Please ensure that each section is presented in a proportional amount of detail without using subheadings. The conclusion is unproportionally short: I would like the authors to make as much effort for this section as for the rest of the manuscript. Please present the abstract with 200 words according to the guidelines of the journal [1], focusing on proportionally presenting the background, results, methods, discussion, and conclusion (without the headings of subsections). The background should include the general background (one to two sentences), the specific background (two to three sentences), and the current issue addressed to this study (one sentence), leading to the objectives. In this subsection, I would like the authors to lay out basic information, a problem statement, and their motivation to break off. The methods should clarify the authors’ approach, such as study design and variables, to solve the problem and/or make progress on the problem. The conclusion should open with one sentence describing the main result using such words like “Here we show”, which should be followed by statements such as the potential and the advance this study has provided in the field and finally a broader perspective (two to three sentences) readily comprehensible to a scientist in any discipline [2–4].
We have extensively edited, structured, and improved the abstract according to your detailed instructions. The abstract now contains 199 words.
- Keywords: Please verify that the listed keywords are included in the MeSH index: Please list ten keywords chosen from Medical Subject Headings (MeSH) [5].
Now all keywords have been carefully selected from Medical Subject Headings.
- Results: I recommend removing all statistical values in the text and present them in the tables. I suggest presenting figures in colors and closing this section with a paragraph that puts the results into a more general context.
All statistical values have been removed from the text. These values are shown in the tables and figures. The figures are now in color. The Results section has been concluded with a closing paragraph to describe the findings in a more general context.
Material and Methods: I recommend opening this section with a short introductory paragraph.
We have provided a short introductory paragraph for this section
- Tables andFigures: According to the journal’s guidelines, please provide a short explanatory caption for the table within the text.
We have provided a short explanatory caption within the text
- References: Please follow the journal’s guidelines [4]. Please list authors’ name with commas and semicolons and use et al. after ten names of authors; abbreviated journal names end with periods; and volume numbers be italicized.
We have carefully followed the journal’s guidelines.
This manuscript is a resubmission of an earlier submission. The following is a list of the peer review reports and author responses from that submission.
Round 1
Reviewer 1 Report
Comments and Suggestions for Authors
Dear Authors,
I found the topic very interesting and your research idea piqued my interest. I agree with you that increased and/or continuous stress can cause liver damage, but unfortunately there are currently few studies dealing with this topic. However, I no longer agree with your presentation method, description of the experiment, and the conclusions. I think that you have a lot to improve on the article, so I reject your article now, but I recommend that you use my comments to improve your article and I support you in resubmitting the article. After that, you'll see a mix of comments on form and content, separated into bullet points:
1. In the case of affiliations, the Petrovsky National Research Center of Surgery lacks the postal code, city and country name.
2. After the affiliations, you should also list the authors' email addresses
3. The description of the abstract is easy to follow and attention-grabbing, but it should be a maximum of 200 words, this abstract is apparently more.
4. The Journal provides a template, it can be seen that it was used, but it can be noticed in several places that the font and style dictated by the template were changed during the insertion. Please use the template correctly, and when pasting the text into the template, use the paste text only option so that you can keep the criteria set by the template.
5. I recommend reconsidering the title, the title describes the results sufficiently, but not concisely.
6. In the text, figures should appear as Figure, not as Fig. The letters a and b in the figure should be lowercase. Figures and tables should appear after the paragraph referring to them. For example, I should display table 1 after line 110.
7. The presentation of the table is inappropriate and difficult to interpret, suggestions: in Tables 1 and 2, put the units of measure in parentheses, use *, #, symbols in the table legend. Also explain the numbers after the plus and minus signs.
8. The display of the p-value is not consistent. For example, p should appear everywhere in lower case.
9. For the table, the table description should appear above the table, the figures and the list of figures should appear below the figures.
10. In the case of the figures, the lines indicating the comparison of the groups and the * indicating the significance levels overlap, so it is difficult to interpret the comparisons. Please improve the figures so that it is better to see which significance level belongs to which comparison.
11. The fig. In the case of 4, the size of the scale bar is difficult to see, in the figure legend, mention the size of the scale bar in the pictures.
12. What would (1.8; 2.5) mean in table 2, why is this an empty row in the table?
13. The figure above line 128 has no figure legend, and between lines 234-236 there is probably a figure legend without a figure.
14. Neither the materials nor the results section defines what is considered a control group.
15. The study description is contradictory, in the materials and methods section it says that 10 control animals and 30 experimental animals, then in the results section it says that the stress experiment was carried out in 20 cases, which were divided into two groups. Table 1 also lists 10 and 20 cases in the LA group.
16. The results are difficult to interpret due to an incomplete experimental description and the methods section of materials. The introduction and the statistical analysis of the results are adequate and satisfy the requested level.
17. In the discussion, the statements are too direct at the beginning, an argument should be built on their results. You explain some sections correctly, but please mention what the limitations of your project were.
18. Why did you only select male individuals and not perform the experiment on female rats?
19. Have you performed protein detection by immunohistochemistry/immunofluorescence or Western blot, which would indicate liver damage or lipid peroxidation, or mitochondrial dysfunction or oxidative stress? Or RNA level analysis?
Reviewer 2 Report
Comments and Suggestions for Authors
The authors used an experimental model of post-traumatic stress disorder (PTSD) to evaluate the effects of chronic stress on the liver of high-anxious (HA) and low-anxious (LA) rats and to search for possible molecular mechanisms involved. They used a predator stress model and concluded that PTSD induces more liver damage in HA rats than in LA rats. This paper discusses the role of glucocorticoids, proinflammatory cytokines, glycogen/glucose and lipid metabolism. They present interesting results, but the work should be improved in some points:
1. The tables use symbols (*, #) that are not described in the footnotes. Please describe the symbols.
2. The abbreviation “NS” in the graphs overlaps the lines.
3. Page 9: the graph has no legend. In fact, there are 8 images/figures and only 7 legends.
4. Describe "PSr".
5. Figure 5 on page 8 shows a Schiff stain for glycogen content, but on page 9 there is a second legend for Figure 5 describing a Sudan III stain for lipid droplets. Please review the content and organization of pages 8 and 9 and improve the presentation.
6. Figure 7: does increased IL-6 increase or decrease oxidative stress?
7. page 11: the authors mention that - "Previously, we reported that chronic PS leads to a decrease in plasma Cort concentration with a concomitant increase in proinflammatory cytokines [24]"-. However, reference 24 does not include a corticosterone determination, the correct article is this: PMID: 32361171; but the latter does not include a cytokine determination, so I think they should cite both reference 24 and PMID: 32361171.
8. Discussion: the authors mention supplementary figure 1 frequently throughout the discussion, I suggest including supplementary figure 1 in the manuscript as the main figure.
9. The authors present a complementary figure 1 with correlations and a color palette. Although the color palette indicates the R-value, or degree of correlation, it is also necessary to know the p-value for each calculated correlation. My question is: do all the correlations shown in figure 1 have a p < 0.05?
10. Page 12: Please describe the "ALT activities".
11. More context is needed in the discussion on small, medium or large mitochondria for better interpretation of the results. Functional differences between them and their implications.
12. The conclusion is on page 15 after the methodology, please check if it is correctly placed.
13. chronic stress is used to increase corticosterone levels, please explain and discuss why the predatory stress model decreases rather than increases corticosterone levels.
Round 2
Reviewer 1 Report
Comments and Suggestions for Authors
Dear Authors,
Thank you for your correction, I think the corrections have greatly increased the value of the article and the manuscript is now recommended for acceptance.